# Helicase Q promotes homology-driven DNA double-strand break repair and prevents tandem duplications

J. A. Kamp [1], B. B. L. G. Lemmens [1], R. J. Romeijn [1], S. C. Changoer[1], R. van Schendel [1] & M. Tijsterman [1,2 ✉]

DNA double-strand breaks are a major threat to cellular survival and genetic integrity. In addition to high fidelity repair, three intrinsically mutagenic DNA break repair routes have been described, *i.e.* single-strand annealing (SSA), polymerase theta-mediated end-joining (TMEJ) and residual ill-defined microhomology-mediated end-joining (MMEJ) activity. Here, we identify *C. elegans* Helicase Q (HELQ-1) as being essential for MMEJ as well as for SSA. We also find HELQ-1 to be crucial for the synthesis-dependent strand annealing (SDSA) mode of homologous recombination (HR). Loss of HELQ-1 leads to increased genome instability: patchwork insertions arise at deletion junctions due to abortive rounds of polymerase theta activity, and tandem duplications spontaneously accumulate in genomes of *helq-1* mutant animals as a result of TMEJ of abrogated HR intermediates. Our work thus implicates HELQ activity for all DSB repair modes guided by complementary base pairs and provides mechanistic insight into mutational signatures common in HR-defective cancers.

[1] Department of Human Genetics, Leiden University Medical Center, Einthovenweg 20, 2333 ZC Leiden, The Netherlands. [2] Institute of Biology Leiden, Leiden University, Sylviusweg 72, 2333 BE Leiden, The Netherlands. ✉email: m.tijsterman@lumc.nl

NA is subject to damage, which can lead to alteration of its nucleotide sequence with potentially detrimental outcomes for an organism. One of the most impactful damages is the breakage of DNA, resulting in the separation of chromosome parts. Traditionally, two mechanisms to repair double-strand breaks (DSBs) were described: non-homologous end-joining (NHEJ) and homologous recombination (HR)[1], which have been intensely studied over the last decades. NHEJ is a conceptually simple mechanism: it directly joins broken chromosome ends together requiring the ligase LIG4 and the DNA-end binding scaffold Ku[2]. While NHEJ can be error-prone, because certain break ends require polishing to allow ligation, it is likely a *grosso modo* accurate DSB repair pathway[3]. HR performs a more complex, multistep reaction that starts with resecting the broken DNA ends in a 5′ to 3′ direction, thus generating a crucial HR intermediate: a 3′ protruding tail[4]. This single-stranded DNA (ssDNA) end, aided by the RAD51 recombinase, can invade a homologous sequence which will serve as a donor for templated DNA synthesis at the 3′ tip. Because the homologous sequence usually is the unbroken sister chromatid, the product of DNA extension is a sequence that is identical to the resected part of the other end of the break, providing the basis for error-free repair: upon dissociation of the extended end from its template, annealing of perfectly complementary stretches of DNA can occur. Depending on the length of the extension and/or the degree of resection, additional gap-filling or trimming needs to occur prior to ligation, which completes a mode of HR repair that is called synthesis-dependent strand annealing (SDSA)[5]. SDSA is thought to account for most HR repair in mitotic cells[6], whereas in meiotic cells DSB repair can also progress via the establishment of double-Holliday junctions that subsequently require additional biochemistry for resolution[7].

Over the years, it has become increasingly clear that NHEJ and HR cannot account for all DSB repair and other modes of repair that prevent chromosomal fragmentation exist[8]. At least two additional pathways have been defined: single-strand annealing (SSA) and alternative end-joining (altEJ)[9,10]. SSA is able to repair DSBs that are flanked by direct repeats (of at least about 50 bp)[9,11,12]. In this mode of repair, end resection exposes the homologous sequences that can directly anneal and seal the break without having to pick up a complementary sequence from the sister chromatid, as is the case in SDSA. In a sense, SSA thus resembles SDSA, yet without the synthesis step. The outcome of SSA is, however, very different from SDSA as the sequence in between the repeats as well as one repeat copy will be lost and SSA is thus intrinsically mutagenic. Apart from mechanistic communalities in SDSA and SSA (e.g. the need for end resection and strand annealing), there are also differences: because direct repeats that guide SSA will rarely be located precisely at the break ends, non-complementary 3′ protruding DNA flaps that inevitably arise upon annealing need to be removed in SSA, a biochemical activity attributed to the structure-specific endonuclease ERCC1/XPF[13]. Because of this necessity, ERCC1/XPF dependency is one frequently used criterium in defining SSA[14], as well as a dependence on the ssDNA binding protein RAD52, which is able to stimulate base pairing of complementary sequences[15,16].

Until recently, altEJ has been the most ill-defined DSB repair pathway. Early work revealed an end-joining activity independent of canonical NHEJ factors[17–19]. Because the cognate repair products were enriched for DNA junctions that appeared to have used small stretches of sequence homology that surrounded the break, the term microhomology-mediated end-joining (MMEJ) has also been used[20]. Recently, it was found that a large proportion of altEJ, yet not all, in many systems involves the action of polymerase theta[21–25]. The term TMEJ for polymerase theta-mediated end-joining was introduced[22] to discriminate between

DSB repair that crucially depends on polymerase theta versus altEJ modes that does not, such as MMEJ in yeast - yeast does not encode polymerase theta-, which was shown to be reliant on the usage of somewhat longer microhomologies (>5 bp)[20]. However, also in species with a polymerase theta ortholog, additional altEJ exist: mammalian cells clearly have altEJ activity that involves longer microhomologies, which is independent of POLQ activity[11]. At present, the underlying mechanism and the genetic requirements of this type of repair is unknown.

Here, we describe the identification of helicase Q (HELQ/HEL308/HELQ-1), a protein previously found to affect DNA repair in various biological systems[26–30], to be essential for polymerase theta-independent MMEJ. We further show that *C. elegans* HELQ-1 is essential for other DSB repair pathways that require the annealing of homologous nucleotide stretches, such as SSA and SDSA. We provide a unified model for HELQ action that explains the redundancies observed in altEJ as well as how the inability of completing HR results in spontaneous accumulation of tandem duplications.

## Results

**Polymerase theta-independent end-joining of G4-induced breaks**. In previous work, we have established that G4 motifs in the *C. elegans* genome provide a source for DNA breaks that depend on alternative end-joining for their repair[31,32]. These motifs, which contain sequential stretches of guanines, can assemble into thermodynamically very stable secondary structures that have the potential to block ongoing DNA replication[33]. In the absence of the G-quadruplex resolving helicase DOG-1/FANCJ, ssDNA gaps form immediately downstream of the replication fork impediment, which persists and are converted to DSBs in the next cell cycle[31] (Fig. 1a). These replication-associated DSBs are predominantly, if not exclusively, repaired by TMEJ, resulting in deletions with a size reflective of the ssDNA gap across the replication impediment[31,32]. Endogenous G4 motifs in *dog-1* deficient *C. elegans* provide a unique model substrate to study TMEJ in vivo because their mutagenic consequences occur at well-defined genomic positions, with a frequency that can be measured with a variety of molecular techniques, including PCR, transgenic and endogenous reporters, and whole-genome sequencing[32,34,35].

Interestingly, while the vast majority of G4-induced DNA breaks are repaired via TMEJ, we found a few cases that are not. In fact, in all genetic contexts in which TMEJ dominates the repair spectrum, such as for structural variations accumulating in *C. elegans* BRCA1 mutant animals[36], we observed a small fraction of repair outcomes that are independent of polymerase theta action. As an example, Fig. 1b presents all genome alterations we identified in *dog-1* and *dog-1 polq-1* mutant animals that were clonally grown for over 120 generations in total after which their genomes were sequenced. While TMEJ products are absent in *polq-1* mutant animals, we found a few larger deletions, which had pronounced microhomology at the deletion junction (Fig. 1c). *C. elegans* TMEJ requires only a minute degree of microhomology —frequently not more than a single base pair[22] and it is suggested that the polymerase action of POLQ-1 extends these microhomologies to promote DSB repair. We hypothesised that more elaborate sequence homology in the flanks of a DNA break may bypass the requirement for polymerase theta to create a thermodynamically stable repair intermediate. For purpose of clarity, in the remainder of the text we will refer to this polymerase theta-independent, extended microhomology-mediated, alternative end-joining mode of repair as eMMEJ. To further investigate the parameters and genetic requirements of eMMEJ, we searched the *C. elegans* genome for loci in which

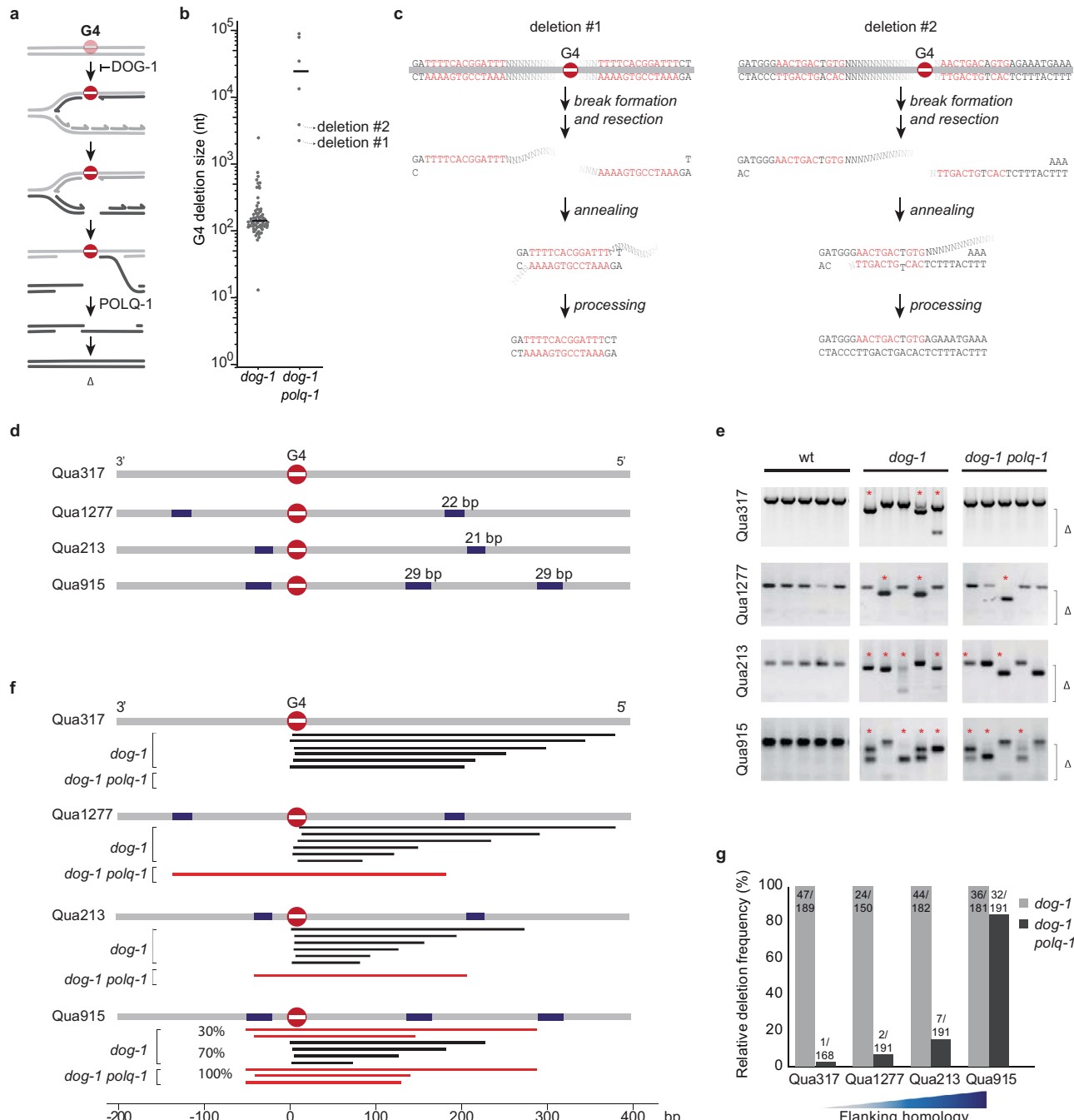

**Fig. 1 Polymerase theta-independent repair of G-quadruplex-induced DSBs is driven by flanking sequence homology. a** Model explaining DSB formation and repair at replication-blocking G-quadruplex structures[31]. **b** Size distribution of deletions that accumulate in the genomes of *dog-1* and *dog-1 polq-1* mutant animals (*dog-1* data from[32]). **c** Potential mechanism explaining the formation of the deletions marked in **b**. **d** Schematic diagram of four endogenous G4 loci with different degrees of flanking homology. The most prominent repeated sequences are indicated in blue. **e** Representative images of PCR-based analysis of the indicated G4-containing loci. Each lane contains the genomic DNA of three adult animals. Asterisks indicate stochastic deletions, which manifest as shorter than wild-type products and Δ indicates the size range of the PCR-amplified deletion products. Deletion sequences are provided as a Source Data file. **f** Graphic illustration of G4 deletions profiles at four endogenous G4 loci. For each locus typical G4 deletions in *dog-1* and *dog-1 polq-1* animals are depicted. Black bars represent homology-independent deletions; red bars represent homology-dependent events. **g** Histogram depicting relative deletion frequencies at the indicated G4 loci as determined by the presence of deletion bands in the PCR-based assay on single worms of *dog-1* (grey bars) and *dog-1 polq-1* mutant animals (black bars). Depicted frequencies are relative to the deletion frequency in *dog-1* single mutants to allow the comparison of loci expressing different stochastic G4 deletion rates. The number of analysed animals is depicted on top of the bars. Gel counts are provided as a Source Data file.

endogenous G4 motifs are flanked by small stretches of sequence homology and selected four loci for in-depth analysis: Qua1277, Qua213 and Qua915, which have an increasing degree of sequence similarity surrounding the G4 motif of about 20–30 base pairs (bps) in size (Fig. 1d), whereas Qua317 is devoid of substantially repeated sequences up- and downstream of the motif, and hence serves as a control. Using a PCR-based approach, we found all four G4 motifs to induce deletions of the TMEJ type in dog-1-deficient animals (Fig. 1e): one deletion junction being located immediately upstream of the G4, the other at varying positions 50 to 300 bps downstream (Fig. 1f). However, while deletion formation at Qua317, which lacks extended micro-homologous sequences (eMH), proved to be almost completely dependent on functional POLQ, deletion formation at Qua1277, Qua213 and Qua915 was less affected: we detected repair products at all of these sites in polq-1 dog-1 mutant animals (Fig. 1e). The dependency of break repair on POLQ was found to be inversely correlated with the degree of homology present in the flanks of the motifs (Fig. 1g). Upon close inspection, we noticed that individual animals (in separately grown populations) often showed deletions of identical size, suggesting a preferred repair outcome specific for each locus (Fig. 1e). Indeed, sequence analysis revealed all TMEJ-independent deletion events to have used eMH flanking the G4 sites (Fig. 2a). Interestingly, the frequency of deletion formation was comparable in POLQ-1 proficient and deficient genetic backgrounds when substantial eMH was present (Fig. 1g). These data support the notion that came forward from the whole genome sequence analysis, i.e. that G4-induced DSBs can be repaired independent of POLQ, but only when the sequence context supports the usage of homologous sequences at either end of the break.

**Helicase Q is essential for eMMEJ of G4-induced breaks**. A logical explanation for stretches of sequence homology able to bypass POLQ is to propose a specific role for POLQ in TMEJ: generating a sufficiently-sized stretch of base pairing (a primer) starting with only a few bases of complementarity at either end of a break. In such a scenario, eMH already present in the immediate vicinity of a DSB makes such activity redundant (Supplementary Fig. 1). However, this explanation does not take into account that POLQ, apart from a polymerase domain, contains a helicase domain that is also required for TMEJ, and which has been suggested to make DSBs accessible for the DNA synthesis step[21,37]. We envisaged that another protein may substitute for this helicase function in eMMEJ, and focused our attention on a logical candidate: Helicase Q (HELQ-1), as this protein's helicase domain is highly similar to the helicase domain within polymerase theta[38].

To address this hypothesis, we crossed a previously characterised mutant allele of helq-1 (tm2134)[30] into dog-1 and dog-1 polq-1 and analyzed G4-induced mutagenesis at the eMH containing locus Qua915 and the eMH lacking locus Qua317. We obtained mutation spectra in which we categorised the obtained deletions by the degree of MH at the junction, and whether or not they contained inserted nucleotides. In HELQ proficient dog-1 animals the Qua915 spectrum is dominated by deletion junctions with eMH (Fig. 2a), whereas a typical TMEJ spectrum is observed for Qua317 (Fig. 2b): minute microhomology at deletion junctions and insertions that, when of sufficient size, can be reliably mapped to flanking sequences.

In perfect support for a causal role of HELQ-1 in eMMEJ, we observed a complete absence of eMMEJ-driven cases at Qua915 in helq-1 dog-1 animals (Fig. 2a); the spectrum now resembles that of Qua317, arguing that these repair products completely rely on TMEJ. Indeed, no deletions were found at Qua915 in helq-1

polq-1 dog-1 animals (Fig. 2c). Conversely, the Qua915 spectrum in polq-1 dog-1 mutant animals is entirely composed of deletions of the eMH type, arguing for a clearly distinct division of labour in altEJ. Together, these data demonstrate that DNA breaks originating from replication-obstructing G-quadruplexes are repaired via TMEJ or via HELQ-1 dependent eMMEJ.

**Helicase Q is a facilitator of TMEJ activity**. Close inspection of the Qua915 deletion spectrum revealed a striking peculiarity: we found that the ratio of (templated) insertions over simple deletions is increased in helq-1 mutant animals when deletions are compared to those derived from HELQ-1 proficient animals (44 versus 23%, respectively, Fig. 2d). One potential explanation is that breaks that are processed to use eMMEJ in HELQ-1 proficient genetic backgrounds have a higher tendency to result in deletions that also contain insertions when HELQ-1 is absent. An alternative explanation is that HELQ-1 is facilitating efficient, non-interrupted TMEJ, and in its absence, one or more successive rounds of abortive TMEJ leads to increased levels of templated insertions (see Supplementary Fig. 1 for a schematic illustration). To discriminate between these explanations, we compared the spectrum of deletions at Qua317, a locus not flanked by obvious eMH, and which is entirely dependent on POLQ-1 for its DSB repair (Fig. 2c). The frequency of insertions at this locus in HELQ-1 proficient animals is 28,6%, similar to the 28% that was previously described for C. elegans G4 sites genome-wide[32]. However, this fraction is considerably higher (48%) in animals that lack HELQ-1 (Fig. 2e). Moreover, also the molecular configuration of the templated insertions derived from helq-1 mutant animals are different: whereas in other genotypes TMEJ products typically contain templated insertions with only one DNA stretch that maps to break-flanking sequences, we found 14% of TMEJ cases in helq-1 mutants to have a "patchwork" appearance, containing multiple stretches of identical sequences (Fig. 2f). We consider these cases products of iterative rounds of primer-template switching during TMEJ, and their increased appearance in helq-1 mutant animals may hint towards an obstacle to TMEJ progression that can be relieved by HELQ-1 activity. HELQ-1 is thus not required for TMEJ but does affect the fidelity of this repair route.

**HELQ-1 is essential for C. elegans single-strand annealing**. The eMMEJ defect in helq-1 mutant animals, as well as the subtler TMEJ phenotype, fits well with a possible role for HELQ in making ssDNA ends available for repair-stimulating base-pairing interaction, e.g. by stripping potentially interfering ssDNA binding proteins. A similar biochemical activity is proposed to facilitate yet another mode of DSB repair, i.e. single-strand annealing (SSA), which has been described as a pathway that repairs DSBs by making use of large stretches of near-identical sequences (>50 bp) up- and downstream of the break site. To assess whether HELQ-1 acts in SSA we used a reporter assay that we previously developed and validated for C. elegans. In this transgenic reporter, schematically illustrated in Fig. 3a, a recognition site of the I-SceI endonuclease is flanked by two large (~250 bps) homologous sequences, which are part of a LacZ gene. The use of these large stretches of homology to repair the I-SceI-induced DSB will lead to the restoration of an otherwise interrupted LacZ open reading frame (Fig. 3a). LacZ expression, quantified by staining animals for B-galactosidase activity, thus serves as a proxy for SSA activity. To optimise the utility of this reporter system, we performed all experiments in an NHEJ (lig-4) deficient genetic background, in which SSA is more prominently detectable (but surprisingly not reliant on the structure-specific endonuclease XPF (Supplementary Fig. 2a)): ~70% of lig-4

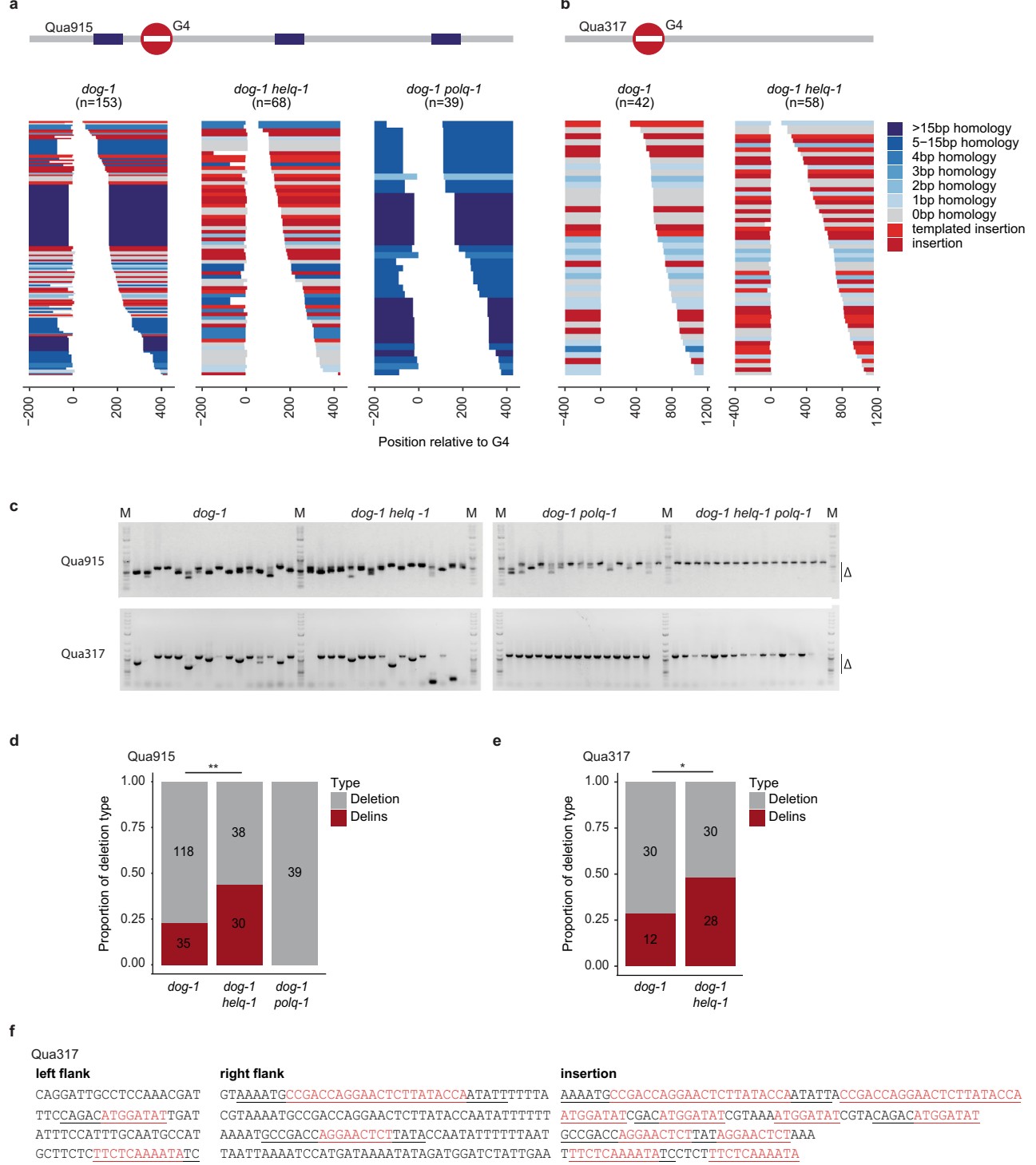

deficient reporter animals display patches of blue cells (Fig. 3b, c). We next assayed *helq-1 lig-4* animals and found a profound decreased number of LacZ positive animals, indicative of impaired SSA (Fig. 3c). It should be noted that a quantification based on the number of animals that display LacZ positive cells severely underestimates the SSA defect in *helq-1 lig-4* animals: while *lig-4* control animals display large segments of LacZ positive cells within individual animals, *helq-1 lig-4* animals typically had only one or two blue cells (Supplementary Fig. 3). Identical staining patterns and intensities were found for *lig-4* and *lig-4 polq-1* animals (Fig. 3c), demonstrating that POLQ-1 (in contrast

to HELQ-1) does not act in SSA, nor suppresses it. From these data we conclude that HELQ-1 is critical for SSA repair of targeted DNA breaks, in addition to being essential for eMMEJ of replication-associated DNA breaks.

**SSA is dominant over TMEJ**. We next considered the fate of DNA breaks that cannot be repaired by SSA in *helq-1* mutant animals: are these now a substrate for TMEJ? To address this question, we PCR-amplified the reporter locus two days after DSB induction at the I-SceI recognition site in order to inspect the DNA for repair outcomes other than SSA products. While some

**Fig. 2 HELQ is essential for eMMEJ of G4-induced DSBs and a facilitator of TMEJ. a** Spectra of deletions occurring at the Qua915 locus, which is schematically depicted above, for the indicated genotype. Single deletion events are piled and sorted from top to bottom by deletion end-point relative to the G4 motif set at 0. Deletion events are colour-coded according to the following mutational classification: grey for simple deletions without apparent MH at the junction, blue for simple deletions with MH at the junction, wherein the saturation level of the blue colour increases with an increasing amount of homology identified. Deletions containing insertions are in red: bright red for insertions that can be reliably mapped to the flank of the deletion, dark red for insertions of which the origin could not be determined with certainty. Deletion sequences are provided as a Source Data file. **b** Spectra of deletions occurring at the Qua317 locus, which is schematically depicted above, for the indicated genotype. Colour coding is identical to (**a**). Deletion sequences are provided as a Source Data file. **c** Representative images of PCR-based analysis of the indicated G4-containing loci for the indicated genotypes. Each well contains the DNA of 10 animals. 2-Log DNA ladders (indicated by 'M') are used as markers for size reference. Uncropped gel pictures are provided as a Source Data file. **d** Proportion of deletion types at Qua915. The difference in the ratio between deletions without and with insertion were tested using the Chi-square test (**$P < 0.01$). **e** Proportion of deletion types at Qua317. The difference in ratio between deletions without and with insertion were tested using the Chi-square test (*$P < 0.05$). **f** Examples of deletions with a complex configuration of templated insertions. For each case, the insertion, as well as the left and right flank of the corresponding deletion, is depicted. Inserted nucleotide stretches that are identical to the flank of the deletion junction are underscored in both the insertion and the cognate flank. Nucleotide stretches present multiple times within one insertion are depicted in red.

deletion products that upon sequencing displayed all hallmarks of TMEJ were found in *lig-4* control animals, their incidence profoundly increased in *lig-4 helq-1* mutant animals (Fig. 3d, e). We confirmed that these deletions were brought about by TMEJ activity as no smaller than wild-type bands were seen in *lig-4 helq-1 polq-1* animals (Fig. 3d). We conclude from this outcome that in cases where profound homology is present, SSA is the preferred pathway for DSB repair and that TMEJ will act as a backup if SSA is compromised. The notion that SSA substrates can be channelled into TMEJ suggests that POLQ can act on DSBs that are resected and likely covered by ssDNA binding proteins, as suggested previously[37].

In addition, and in agreement with our observations at G-quadruplex-induced DSBs, also at endonuclease-induced DSBs, we find evidence for (i) HELQ being essential for eMMEJ and (ii) having a facilitating role in TMEJ. On the first note, we observed residual deletion mutagenesis in *lig-4 polq-1* double mutant animals but not in *lig-4 polq-1 helq-1* triple mutant animals (Fig. 3d); and also here we find HELQ-dependent repair products to be characterised by eMH at the junctions (Fig. 3f). On the second note, we once more observed a markedly higher percentage of (templated) insertions in the HELQ deficient genetic background: nearly half of all deletions (34 out of 71) contain an insertion (Fig. 3f and Supplementary Fig. 3). Of these, 19 can be classified as templated insertions, including three which have complex, repetitive configurations (Supplementary Fig. 3) similar to the patchwork insertions observed at G4 sites (Fig. 2e).

**Helicase Q is essential for synthesis-dependent strand annealing.** Having identified an essential role for HELQ-1 in two mutagenic DSB repair pathways, i.e. eMMEJ and SSA, which both involve annealing of complementary bases, we next wished to investigate whether HELQ-1 acts in error-free repair. In particular, it seems logical to assay the sub-pathway of homologous recombination that also requires an annealing step: SDSA. In a late step of SDSA, two single-stranded overhangs, that are complementary in sequence resulting from extending one end of the break using the sister chromatid as a template, anneal to finalise repair without the loss of DNA sequence (see Fig. 4a for a visual representation). To test the potential involvement of HELQ-1 in SDSA, we made use of a previously described DSB-repair reporter that can read out HR in somatic cells of individual animals[39]. This reporter, schematically illustrated in Fig. 4b, consists of a GFP expression cassette but having the GFP open reading frame (ORF) disrupted by an I-SceI recognition site. Downstream of this cassette, a DNA segment has been engineered that has the disrupted part of the GFP ORF as well as up- and downstream sequences, but by itself does not encode for a functional GFP protein. This downstream DNA segment can be used as a

template for the repair of a DSB introduced by the I-SceI endonuclease within the corrupted GFP, eventually leading to restoration of the GFP ORF. Reporter animals thus express GFP in cells in which HR repair took place[36,39]. We previously established that this reporter monitors BRCA1/BRC-1 dependent SDSA[36] (double holiday junction-mediated HR in worms is independent of BRC-1[40,41]). Figure 4c displays a severe reduction in GFP ORF correction that results from HELQ-1 loss. Similar to *brc-1* deficiency, loss of *helq-1* causes a significant reduction in GFP ORF correction, indicative of a severe defect in SDSA. Despite a comparable reduction in SDSA for *brc-1* and *helq-1* mutants, the epistatic analysis argues also for non-overlapping roles of BRCA1/BARD1 and HELQ in DSB repair: GFP correction is further reduced in animals double deficient for HELQ-1 and BARD1/BRD-1 (Supplementary Fig. 2b), the obligatory binding partner of BRCA1/BRC-1[42]. In support of this notion, we found that *helq-1 brd-1* double mutant animals are also more sensitive to DSB-inducing radiation than either of the single mutants, which by themselves are more sensitive to radiation than wild-type animals (Supplementary Fig. 2c).

To further validate HELQ-1 action in DSB repair that includes a DNA synthesis and annealing steps (as is SDSA), we used an independent method[43] in which templated repair of CRISPR/Cas9-induced breaks can be assessed by measuring gene correction using single-stranded oligodeoxynucleotides (ssODNs). In this approach, schematically illustrated in Fig. 4d, animals are injected with plasmids that drive Cas9 expression and a sgRNA that targets the endogenous *dpy-10* locus, together with a short (101 nucleotides) ssODN containing a specific *dpy-10* mutation. This base pair mutation when introduced into the genome leads to a morphological change that is distinguishable from *dpy-10* loss of function-inducing sequence resulting from mutagenic end-joining[44,45]. Animals that based on their morphological phenotype had an altered *dpy-10* locus were isolated. Subsequent sequencing analysis revealed that in repair proficient controls ~63% of the animals were altered via ssODN-mediated gene correction. In contrast, in *helq-1* mutant animals, only 8% of the repair proved ssODN-mediated (Fig. 4f), hence strengthening a fundamental role for HELQ-1 in promoting SDSA. In line with the notion that TMEJ is responsible for mutagenic end-joining in the *C. elegans* germline[46], we exclusively found ssODN-mediated events in *polq-1* mutant animals. No altered alleles were obtained in *helq-1 polq-1* double mutant animals, thus excluding other mutagenic repair pathways acting on these CRISPR-induced DSBs.

**Helicase Q prevents tandem duplications.** We next investigated whether HELQ-1 loss affects genomic integrity also during non-challenged growth, given that animal growth and development appears unaffected in *helq-1* mutants[47]. Previously, we found that

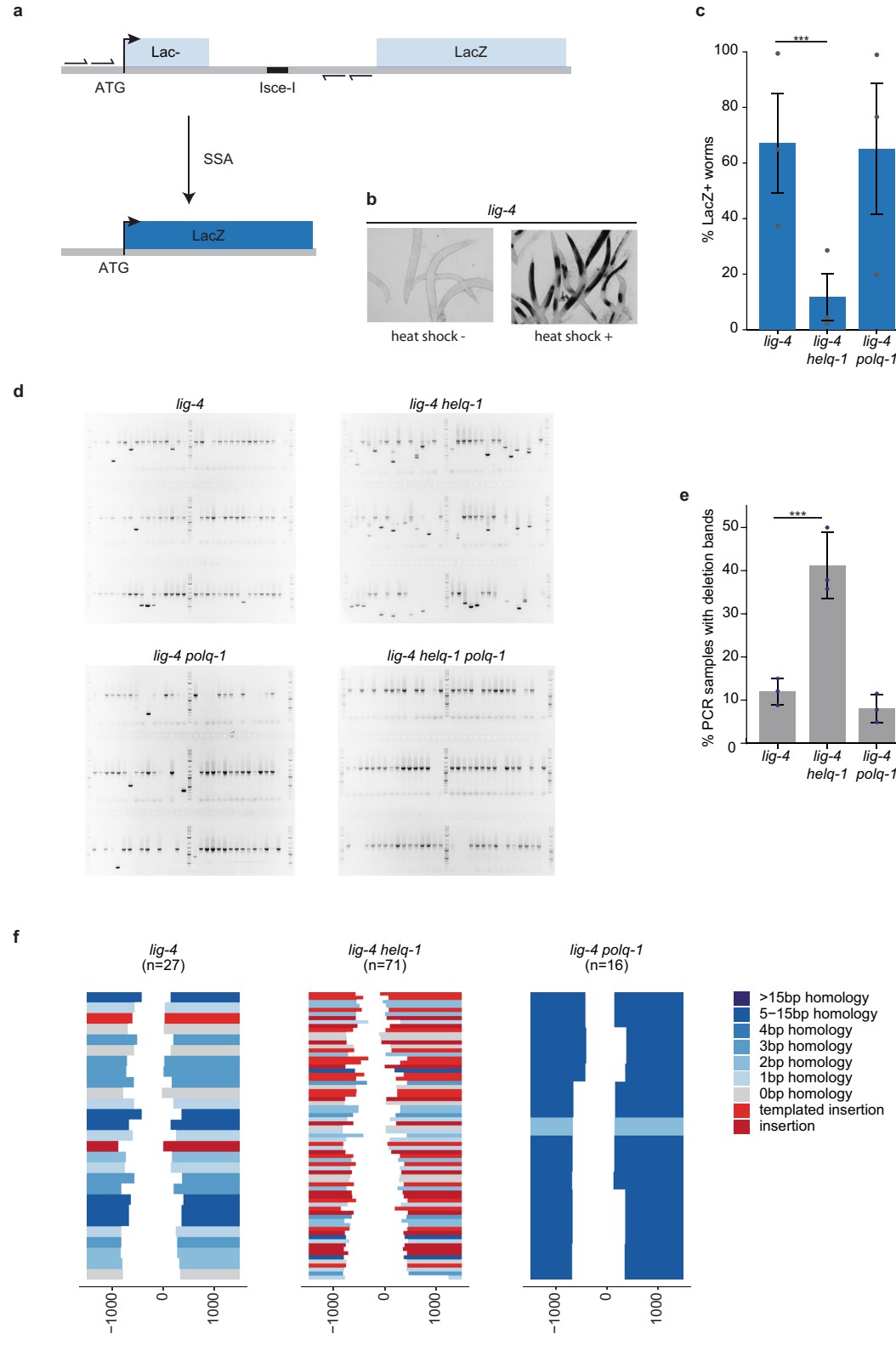

**f**

lig-4 (n=27)    lig-4 helq-1 (n=71)    lig-4 polq-1 (n=16)

>15bp homology
5–15bp homology
4bp homology
3bp homology
2bp homology
1bp homology
0bp homology
templated insertion
insertion

−1000   0   1000    −1000   0   1000    −1000   0   1000

Position relative to DSB

*brc-1* and *brd-1* mutant animals, which despite having impaired HR also grow and develop similar to wild-type animals, accumulate structural variants in their genomes[36]. This increased level of mutagenesis argues that SDSA plays a role in the maintenance of genetic integrity, likely through error-free repair of DSBs that may be of meiotic origin or spontaneously arise during proliferation. To monitor mutagenesis at a genome-wide scale and in

an unbiased fashion we established independent mutation accumulation lines: we separately grew populations of *helq-1* deficient animals for 50 generations after which we performed whole-genome sequencing (WGS). For this analysis, we used a previously described null allele of *helq-1* as well as a newly-made knockout generated by CRISPR. Strikingly, for both mutant strains, we found mainly one specific type of genome alterations

**Fig. 3 SSA deficiency in *C. elegans* lacking HELQ-1. a** Schematic representation of the single-strand annealing (SSA) reporter[14]. A DSB can be introduced at the I-SceI recognition site by the I-SceI endonuclease. DSB repair by annealing of ~250 bp region of sequence identity placed up and downstream of the I-SceI recognition site leads to a functioning LacZ open reading frame. Small arrows depict primers used in the PCR analysis. **b** Representative pictures of *lig-4* mutant animals carrying the reporter transgenes that were heat-shocked or mock-treated to induce I-SceI expression, followed by staining for B-galactosidase expression. **c** Histogram depicting the percentage of LacZ positive worms for the indicated genotype. Experiments were performed in triplicate (***$P < 0.01$; Chi-square test). Each dot represents the average percentage of each replicate. Error bars represent SEM. Staining quantifications are provided as a Source Data file. **d** Representative images of PCR-based analysis of the reporter locus at the I-SceI site for the indicated genotypes. Each well contains the DNA of one animal. Wild-type bands are expected because many cells in the animal are insensitive to heat shock-driven I-SceI expression. 2-Log DNA ladders are used as markers for size reference. **e** Histogram depicting the percentage of PCR samples containing a deletion. Each dot represents the percentage of deletions found in 96 samples. Experiments were performed in triplicate (***$P < 0.01$; Chi-square test). Error bars represent SEM. Deletion counts are provided as a Source Data file. **f** Spectra of deletions occurring at the I-SceI site of the SSA reporter for the indicated genotype. Single deletion events are piled and sorted from top to bottom by deletion end-point relative to the I-SceI cut site set at 0. Deletion events are colour-coded according to the following mutational classification: grey for simple deletions without apparent MH at the junction, blue for simple deletions with MH at the junction, wherein the saturation level of the blue colour increases with an increasing amount of homology identified. Deletions containing insertions are in red: bright red for insertions that can be reliably mapped to the flank of the deletion, dark red for insertions of which the origin could not be determined with certainty. Deletion sequences are provided as a Source Data file.

accumulating, i.e. tandem duplications (TDs) (Fig. 4f), corroborating recent data describing mutational signatures in *C. elegans* DNA repair mutants[48,49]. These TDs, which are rarely found in wild-type nematodes[36,49], range in size from ~100 bp to 10 kb, with a median of ~1 kb. Importantly, the DNA junctions that arise between the two direct repeats show all hallmarks of TMEJ: overrepresentation of microhomology and the occasional presence of templated insertions (Supplementary Fig. 4). This outcome fits a model in which TMEJ acts on failed SDSA intermediates: in the absence of HELQ-1, extended DNA ends (using a sister homologue as a template) will fail to anneal to the complementary sequence of the other break-end. Direct end-joining of both ends now results in a duplication of the sequence that was copied during the synthesis step of SDSA (Fig. 4g). Indeed, TD formation in *helq-1* deficient conditions proved to be dependent on functional POLQ-1 (Fig. 4f). We find the combined loss of *helq-1* and *polq-1* to only mildly affect population growth, which may not be surprising given the low rate in which TDs accumulate in *helq-1* mutants under non-stressed conditions (i.e. 0.1–0.2 per animal generation). Increasing the genomic damage load by exposing animals to ionising radiation indeed reveals redundancy for these proteins in the cellular response to DNA damage (Supplementary Fig. 2d).

## Discussion
Here, by using validated transgenic DSB-reporter animals and mutational footprint analysis at G-quadruplex and CRISPR-induced DSBs, we show in *C. elegans* that HELQ-1 is necessary for all DSB-repair mechanisms that are guided by annealing of extensive stretches of complementary bases at break ends.

HELQ is an ATP-dependent 3′-5′ DNA helicase of the HEL308 family that is present in archaea and eukaryotes[38,50]. Loss of HELQ in mouse or human cells causes germ cell attrition and ICL sensitivity[28,29,51]. Because of persistent RAD51 foci in *C. elegans*[47] and mammalian cells[28] lacking HELQ, a post-synaptic role in HR was proposed. More recently it was found that HELQ acts redundantly with a pathway involving the proteins HROB, MCM8, and MCM9—cells lacking both HROB and HELQ had severely impaired HR—leading to the suggestion that HELQ could act in parallel to these proteins in HR to promote bubble migration after D-loop establishment[52]. However, our finding that HELQ also acts in SSA and MMEJ -pathways that do not entail the processing of D-loops—points to a role in DSB repair that is shared between these pathways.

In this respect, it is noteworthy that the annealing step in SDSA is conceptually indistinguishable from that in SSA: perfect alignment of large stretches of complementary nucleotides that are present at the ends of a break in an ssDNA configuration. Early work already

pointed towards a role for HELQ in SSA in *Drosophila*[53], although different reporter systems have led to seemingly incompatible data[27], likely because of different experimental features such as the distance between, and the length of, the stretch of homologous sequence flanking a DSB. SSA is most well-studied in yeast which, however, does not encode HelQ, and where *RAD52* and its paralogue *RAD59* have shown to be required[54]. While RAD52 also promotes SSA in mammalian cells, there is strong experimental support for the existence of additional, partly redundant annealing factors[11,55]. For instance, RAD52 inactivation in U2OS cells reduced SSA at CRISPR-induced breaks to approximately 50% of SSA in RAD52 proficient cells[11]. *C. elegans* does not encode a RAD52 ortholog and the absolute requirement for HELQ-1 in *C. elegans* SSA further supports this notion of functional redundancy. However, it may well be that only HELQ promotes MMEJ. While we demonstrate this here for worms, recent work in human cells revealed that repair using stretches of homology of about 20 bp did not require RAD52, nor polymerase theta[11], thus directly pointing towards other genetic requirements for polymerase theta-independent altEJ.

Our data provoke the idea that HELQ supports a wide range of MH usage, in fact arguing that a strict separation between MMEJ and SSA is not sustainable as the biochemistry and genetic requirements may be identical. Yet the usage of this mode of repair is being dictated by the kinetics potentially as a function of MH length and/or strength of the base-pairing interaction. In DSB contexts where annealing results in 3′ flaps that need to be processed before gap filling can occur, an additional genetic requirement for a structure-specific endonuclease is logical (hence the need for XPF in many experiments addressing SSA).

With respect to usage of HELQ-dependent MMEJ/SSA over TMEJ, it is of interest to note that the lower limit of HELQ-dependent MMEJ (>5 bp) is overlapping with TMEJ, and it was recently shown that somewhat larger microhomologies positively influence TMEJ efficacy, making the reaction less prone to abortive synthesis[51]. As previously proposed[11,36,56], the sole function of polymerase theta in alternative end-joining may thus be to convert minimal MH (1–5 bp) into larger stretches that have sufficient thermodynamic stability and priming potential for error-free DNA synthesis, e.g. by Pol delta aided by other helicases such as HELQ. In case such a 'primer' can be generated by base pairing of small complementary stretches of sequence, the need for POLQ in altEJ becomes obsolete[11], as evident in our experiments as well as in mammalian cell culture work[11] (Figs. 1–3).

The mechanism by which HELQ makes ssDNA available for base-pairing interactions is currently unknown, but primarily because of recent in vitro work we favour a function in removing potentially annealing-interfering ssDNA binding proteins such as

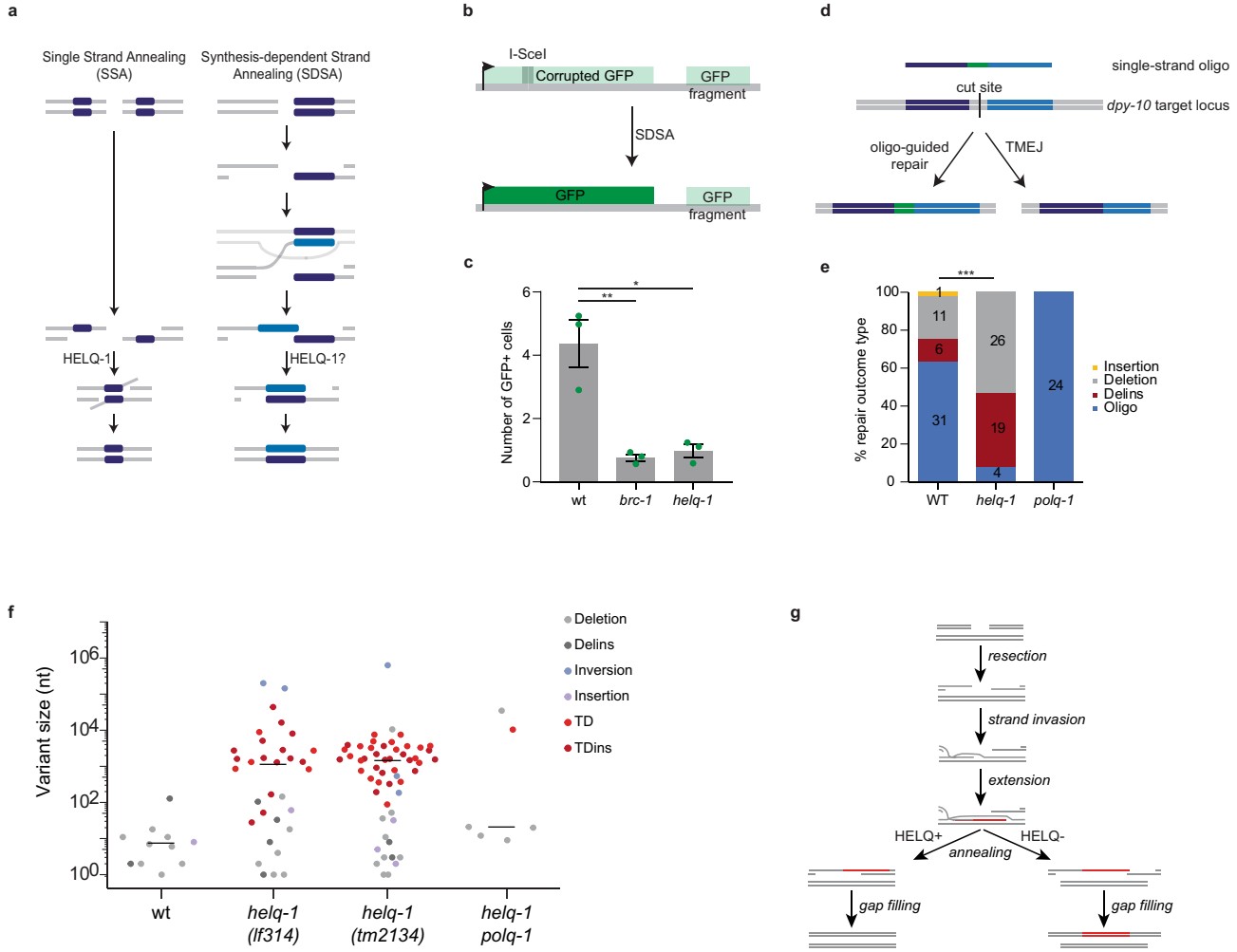

**Fig. 4 HELQ-1 acts in synthesis-dependent strand annealing and suppresses tandem duplications. a** Models for SSA and SDSA repair of DSBs. Homologous stretches of DNA are depicted in dark blue. **b** Schematic representation of the SDSA reporter[39]. A DSB can be introduced by the I-SceI endonuclease at its recognition site positioned within a corrupted GFP expression cassette. SDSA repair using a GFP segment cloned downstream of the expression cassette will restore the GFP ORF leading to GFP expressing animals. **c** Histograms depicting the average number of GFP expressing cells per worm for the indicated genotype. Experiments are performed in triplicate (*P < 0.05, **P < 0.01; two-sided t-tests). P value WT vs. brc-1: 0.0085. P value WT vs. helq-1: 0.0115. Each dot represents the average number of each replicate. Error bars represent SEM. Quantifications are provided as a Source Data file. **d** Schematic representation of the CRISPR/Cas9-induced gene correction assay. CRISPR-induced DSB at the dpy-10 locus can be repaired via end-joining or using an injected single-stranded oligo as a template. The sequence of the oligo differs from the target locus at the centre (in green). Mutation sequences are provided as a Source Data file. **e** Histogram depicting the different categories of repair outcomes. Insertions are depicted in yellow, simple deletions in grey, deletions containing insertion in red, outcomes of repair using the oligo are depicted in blue. The difference in ratio between ssODN-guided repair and indels were tested using Chi-square test (***P < 0.001). **f** Size distribution of structural variations that accumulate in the genomes of the indicated genotype (wild-type data derived from ref. [46]). Mutation sequences are provided as a Source Data file. **g** Mechanistic model for the aetiology of tandem duplications (TDs) in helq-1 mutant genetic backgrounds.

RPA or RAD51. HELQ, purified from an archaeal or human source, was found to interact with RPA and able to modulate the RPA-DNA binding including its displacement from ssDNA[57,58]; C. elegans HELQ-1 was shown to promote the disassembly of RAD51 from DNA in vitro[47]. Also, the helicase domain in POLQ, which is highly similar to HELQ, was capable of removing RPA from resected DSB ends in vitro to allow their annealing[37]. Furthermore, the increased incidence of templated insertions (occasionally having more complex sequence configurations) in HELQ-1 deficient contexts suggests an obstacle to DNA synthesis leading to multiple abortive rounds of TMEJ. It is tempting to speculate that TMEJ can initiate DNA synthesis using minute MH available at the terminal ends of a DSB[56,59] yet requires helicase activity (within polymerase theta itself or borrowed from HELQ) for

removing ssDNA binding proteins, as suggested[37]. One way to explain how HELQ suppresses templated insertions is to envision two possible repair routes acting on DSB ends that result from abortive TMEJ, i.e. eMMEJ and (another round of) TMEJ. Aborted TMEJ can produce 3′ tails that now contain stretches of eMH, because in TMEJ one 3′ ssDNA end is extended using the other end as a template. Repair of such intermediates via HELQ-mediated eMMEJ will result in footprints in which the aborted reaction escapes detection. However, a new round of TMEJ that uses the outermost nucleotides of opposing break ends will then produce a templated insertion. According to this rationale, the frequency of templated insertions will increase in a HELQ deficient genetic background. Of particular interest in this respect, TMEJ outcomes in plants, which do not encode HELQ (while encoding

RAD52 and POLQ), are far more complex than those observed in *C. elegans*, the majority having templated insertions, which very frequently have a patchwork configuration[24,60].

Our data also allow us to address hierarchy in DSB-repair pathway usage. In a direct comparison, HELQ-mediated SSA was shown to dominate over TMEJ and loss of SSA in *helq-1* mutants coincided with more frequent TMEJ products at induced DSB. This refunneling halfway during SSA repair suggests that TMEJ can act on resected DSB ends, likely covered by ssDNA binding proteins, as already discussed above. It also suggests that TMEJ functions as a backup for annealing, which is in line with previous findings that TMEJ can compensate for impaired HR processes[21,23,36,56,61]. Indeed, a further testimony for TMEJ's ability to act on HR substrate is the spontaneous accumulation of TDs in the genomes of HELQ-1 deficient animals when grown in unchallenged conditions—a phenotype previously linked to the HR deficiency in BRCA1 mutant cells[62]. Also in *C. elegans* BRCA1 deficiency leads to the accumulation of TDs, and we have previously suggested that impaired annealing of an extended strand during SDSA (as a consequence of impaired resection) underlies this phenotype[36]. Similarly, we suggest that TDs arise in HELQ-1 deficient animals because of an inability to complete a downstream step in SDSA: the annealing of an extended break-end after being released from its template within the D-loop. Instead, this extended end is (end-)joined to the other DSB end by TMEJ resulting in a TD, a signature motif in certain cancer genomes.

In conclusion, we demonstrate that HELQ-1 is needed for DSB repair that involves the annealing of complementary nucleotides, making it an essential factor in SSA, MMEJ and SDSA. We find that altEJ in worms is the sum of two pathways, both requiring a HELQ-family helicase: HELQ in eMMEJ and POLQ in TMEJ. AltEJ may be the rescue for chromosomal breaks that cannot be repaired by the predominantly error-free pathways NHEJ and HR. While perhaps not being as critical for cell survival as NHEJ or HR, altEJ pathways are intrinsically mutagenic and may thus be the primary cause for DSB-induced genomic alterations[36,46,63].

## Methods

**C. elegans genetics**. Nematodes were cultured on standard NGM plates seeded with OP50 bacteria at 20 °C[64]. The following alleles were used in this study: *brc-1(lf288)*, *dog-1(gk10)*, *helq-1(tm2134)*, *lig-4(ok716)* and *polq-1(tm2026)*. Using CRISPR/Cas9 mutagenesis a novel *helq-1* allele (*lf314*) was generated (guide RNA sequence 5′-GCAGATTTGCACCTTCGTAT-3′) by injecting plasmids in germlines of N2 Bristol worms using standard *C. elegans* microinjection procedures. The injected plasmids encode Cas9 protein (pDD162 (Peft-3::Cas9, Addgene 47549)), the guide RNA (pMB70 (pJJR50)) and phenotypic markers (mCherry (pCFJ90, pGH8 and pCFJ104) and rol (pRF4)) to identify transgenic F1 progeny.

**Mutation accumulation assays**. Mutation accumulation lines were generated by cloning out ten F1 animals from one hermaphrodite. Three nematodes were transferred to new plates each generation. MA lines were maintained for 40–60 generations (Supplementary Table 1). Single animals from the last generation were transferred to new NGM plates. When sufficient offspring was present on these plates, nematodes were washed off with water and incubated for 2 h while shaking to remove bacteria from the intestine. Genomic DNA was isolated using a Blood and Tissue Culture Kit (Qiagen). DNA was sequenced on an Illumina HiSeq platform (2 × 150 bp paired-end reads). The median sequencing depth of the samples is presented in Supplementary Table 1.

**Bioinformatic analysis**. Image analysis, base calling and error calibration were performed using standard Illumina software. BCL output from the HiSeqX and Novaseq6000 platform was converted using bcl2fastq tool. Raw reads were mapped to the *C. elegans* reference genome (Wormbase release 235) by BWA[65] and SAMtools[66]. Pindel[67] was used to call G4-induced deletions. Pindel, Manta[68] and GRIDSS[69] were used for calling structural variations. Variations were considered true if they were covered by both forward and reverse reads and supported by at least five reads. Events were only considered if they were uniquely present in one of the samples. All events were inspected by IGV[70] to ensure the correctness of the call.

**PCR-based assays to identify deletions**. Deletion formation at endogenous G4 DNA loci and at the SSA reporter was assayed using a PCR-based approach. Genomic DNA was isolated either from single worms or pools of worms and subjected to nested rounds of PCRs (see Supplementary Table 2 for primer sequences). L4 stage animals were lysed in 15 μl lysis buffer (50 mM KCl, 10 mM Tris-HCl pH 8.3, 2.5 mM MgCl$_2$.6H$_2$O, 0.45% NP-40 (IGEPAL CA 630), 0.45% Tween-20) and 1 μl lysate was transferred into 15 μl external PCR mix. After the external PCR, ~0.1 μl external PCR mix was transferred to internal PCR mix using a 384 pin replicator (Genetix X5050) for the internal PCR. Deletion products were discriminated based on size by gel-electrophoresis using a 2-log DNA ladder (New England Biolabs N3200S). Repair footprints were analysed using Sanger sequencing. Uncropped blots are provided as a Source Data file.

**Sanger sequence analysis**. A custom Sanger Sequence-analyser was written (available upon request) to determine sequence alterations at the endogenous G4 loci, SSA reporter and CRISPR target site. Each Sanger sequence was filtered prior to comparison with the reference sequence on the following criteria: a stretch of ≥40 nt was present where each base had an error probability of <0.05 surrounding the break site. All other nucleotides were masked. The filtered high-quality sequence was then compared to the reference sequence and classified as insertion, deletion, wild-type or delins (deletion with insertion).

**Ionising radiation sensitivity assay**. L4 stage nematodes were exposed to ionising irradiation or mock-treated. Per experimental condition, three-seeded NGM plates containing three nematodes were prepared. The irradiated nematodes were removed from the plate after 48 h of egg-laying. The number of hatched and unhatched progeny was quantified 24 h after removal.

**Reporter assays**. SSA was read out using a LacZ reporter[14] and SDSA was read out using a GFP reporter[39] using methods described previously. *C.elegans* populations carrying a reporter were synchronised by incubating the worms in a 3:2 mixture of hypochlorite (Acros Organics) to 4 M NaOH until they were dissolved and only eggs remained. The eggs were washed with M9 buffer (22 mM KH$_2$PO$_4$, 42 mM Na$_2$HPO$_4$, 86 mM NaCl, 1 mM MgSO$_4$) to remove the residual bleach. Eggs were allowed to hatch overnight in M9 buffer. After hatching, larvae were plated on OP50 seeded NGM plates and incubated at 34 °C for 2 h (heat shock). One day after heat shock, I-SceI-mCherry expression was verified using a Leica DM6000 microscope. For SDSA experiments, nematodes were scored for GFP-positive intestinal nuclei using a Leica DM6000 microscope 72 h after heat shock. For SSA experiments, single worms were fixated and stained in 5% X-gal or lysed for PCR analysis two days after heat shock. One hour prior LacZ staining, young adults were heat-shocked at 34 °C for 120 min to induce SSA reporter expression.

**CRISPR/Cas9 experiment**. One day prior to injection, L4 animals were incubated at 15 degrees overnight. Gonads of young adults were injected with a mix containing 20 ng/μl Cas9 (pDD162 (Peft-3::Cas9, Addgene 47549)), 20 ng/μl mCherry marker (pCFJ90 and pCFJ104), 20 ng/μl pBluescript and 20 ng/μl *dpy-10* sgRNA coding plasmid (pMB70 (u6::sgRNA) and 20 ng/μl ssODN described previously[43]. Three days after injecting P0 young adults, F1 progeny were phenotypically screened and phenotypically altered animals (dpy, rol or mCherry phenotype) were transferred to fresh plates. F2 animals were lysed for PCR analysis (primers: FW: 5′-caacgaactattcgcgtcag-3′, RV: 5′-gtggtggctcacgaacttg-3′).

**Reporting summary**. Further information on research design is available in the Nature Research Reporting Summary linked to this article.

## Data availability
The data that support this study are available from the corresponding author upon reasonable request. The raw sequence data generated in this study have been deposited in the NCBI SRA database under accession code PRJNA718436. The N2 wild-type and *dog-1* sequence data were published previously and can be found at NCBI SRA (accession codes PRJNA260487 and PRJNA225882). We used the *C. elegans* reference genome (Wormbase release 235 [https://wormbase.org/about/wormbase_release_WS235#10—10]) in this study. Source data are provided with this paper.

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

## Acknowledgements

Some strains were provided by the CGC, which is funded by the NIH Office of Research Infrastructure Programs (P40 OD010440). This work was funded by an ALW OPEN grant (OP.393) from The Netherlands Organization for Scientific Research for Earth and Life Sciences to M.T.

## Author contributions

J.A.K., B.B.L.G.L. and M.T. conceived and designed the study. J.A.K., B.B.L.G.L., R.J.R., R.v.S. and S.C.C. performed the experiments. R.v.S. performed the bioinformatical analyses. All authors interpreted the experimental results. J.A.K. and M.T. wrote the manuscript.

## Competing interests

The authors declare no competing interests.
