## [Peer Review File · Nature Communications]

Helicase Q promotes homology-driven DNA double-strand break repair and prevents tandem duplication mutationsReviewers' Comments:

Reviewer #1:

Remarks to the Author:

Kamp et al argue for a significant role for Helq in every DSB repair pathway except NHEJ. A large number of elegant assays generate some striking data, which taken together mostly support the argument above. It's a mostly beautiful piece of work, and will be of wide interest.

Major concerns that should be addressed:

- 1) Most importantly, the authors are missing a good argument for gross biological significance. For example, they describe dramatic reductions in aberrations (error prone repair) in double and triple mutants (e.g. *helq+polq*, *helq+polq+lig4*) – is there a compensating loss of viability? If not, can they at least show doubles or triples are hypersensitive to damaging agents?
- 2) The genetic argument for a role for *helq* in Polq dependent and independent repair by TMEJ/MMEJ is good. The authors should perform similar epistasis analysis for SSA (i.e. *helq* with *xpf*) and SDSA (i.e. *helq* with *brc1*).
- 3) Regarding sites devoid of large microhomologies; the authors initially describe Qua375 as such a site, then in line 155, it is Qua317. Are these different sites (then why use different ones?), or is one a typo?
- 4) In Figure 1g, please note the number of deletions scored for each G4 site in the figure (i.e. how many deletions is 100%?; $n > 360$ is too vague).
- 5) Line 109, citation 31; please elaborate.
- 6) Line 183; please quantify what is meant by more frequent "patchwork appearance", or omit.
- 7) Line 336; "As previously proposed..." please also cite Jeremy's work (ref 11) arguing POLQ suppresses a need for larger microhomologies, as well as PMID: 32234782 showing larger microhomologies make POLQ less prone to abortive synthesis/templated insertions. Similarly Line 341; please again cite ref 11 that larger microhomologies eventually make POLQ dispensable for a-EJ in mammals.
- 8) In the introduction and discussion, please discuss related work in flies (PMC1667076, PMC2013711). The latter is especially relevant, as it argues against a role for HELQ in SSA in flies.

Grammar and style suggestions.

- 1) Line 55: the manner in which "decent" describes "size" is at best vague, and likely to be confusing to many readers.
- 2) Line 174: "One potential explanation..." has a comma splice that makes the sentence difficult to understand.
- 3) Line 234: "...and in perfect agreement...", and line 291 "...perfectly fits...". Suggested that the authors will inspire more agreement amongst readers without offering opinions as to "perfection".
- 4) Line 310 "...besides SDSA..." this clause makes the sentence difficult to parse.
- 5) Line 317 "...in part..." suggest change to "partly"
- 6) Line 319 "...only to about..." this clause makes the sentence difficult to understand.

Dale Ramsden

Reviewer #2:

Remarks to the Author:

This is a review of NCOMMS-21-13096, which is an elegant study of the influence of the gene HELQ on chromosomal double strand break (DSB) repair and genome stability in *C. elegans* (worms). Importantly, even beyond the mechanistic findings with HELQ per se, the classes of phenotypes observed in HELQ- worms provides significant insight into the classes of backup chromosomal break repair pathways that exist in this organism. It is also a significant study, since the factors that mediate MMEJ/SSA beyond POLQ and RAD52 have remained poorly understood. The study is elegant in that it

uses several complementary reporter assays involving distinct types of DSBs: spontaneous events caused by G4 and loss of FANCDJ, as well as I-SceI and CAS9 DSBs. Several classes of DSB repair events are found to be suppressed/promoted by HELQ that point to a role in events requiring use of longer stretched of homology, which are contrasted from those promoted by DNA Polymerase theta. The manuscript is also written in a compelling manner. For two of the results, however, I would request additional details of the findings, either in the main figure or in supplemental.

1. In Fig 3f, there is a striking increase in templated insertions / insertions in the lig-4/helq-1 worms. Some examples of these insertions and/or some classification of at least the most common types of insertions will stimulate research in this area. It might also be useful to speculate on the meaning of this result. For examples, this result may point to a role for HELQ in suppressing insertional mutagenesis / suppression of translesion polymerases, and/or it may be that HELQ is important to unwind terminal DSB ends, and in the absence of this activity, polymerases generate ligatable intermediates.

2. For the tandem duplications found in Fig 4f, I have a similar comment. Could the authors show some examples of these TDs? How long is the homology involved in the TDs? Finally, some discussion of Ralph Scully's work on how deregulation of end resection can result in long-tract gene conversion may be warranted here, but of course only if the data on the TDs here are informed by that literature. In summary, more description / diagrams of the nature of the TDs will also increase the impact of the study.

Reviewer #3:

Remarks to the Author:

In this paper, Kamp et al. investigate the source of residual microhomology-mediated end joining that occurs in *C. elegans* polq-1 mutants. They identify the DNA helicase HELQ as being essential for MMEJ repair of G4-related DSBs with flanking microhomologies of greater than 20 nts (which they call eMMEJ). In addition, they show that HELQ facilitates TMEJ and in its absence, repair junctions have larger and more complex insertions. Using several reporter assays, they establish that HELQ-1 is needed for SSA in a lig4 background and for efficient SDSA. In its absence, tandem duplications accumulate in the genome during mutation accumulation experiments. Overall, their results suggest an important role for HELQ in double-strand break repair that involves annealing of complementary ssDNA molecules greater than 20 nt in length.

The novelty of this study lies in its identification of a protein important for the small fraction of end-joining repair events that occur in the absence of pol theta. These have previously been observed in worms, flies, and mouse cells, but their etiology has remained unknown. The involvement of HELQ in SSA and SDSA has been suggested in other systems, but this work brings these observations together under the umbrella of a postulated role of HELQ-1 in removing inhibitory proteins from ssDNA prior to completion of break repair. The data are clear and the figures nicely convey the main points. While I have some questions about the interpretation of some of the data, I think that the study as a whole provides significant insight into DSB repair and pathway choice in *C. elegans* (but perhaps not in other metazoans, see below).

Points to consider:

1. It isn't clear why different control sequences are used for Figure 1 (Qua375) and Figure 2 (Qua317). It would be useful to include sequencing data for the Qua317 locus in the different mutant backgrounds, or alternatively show the sequencing data in helq-1 and dog-1 polq-1 helq-1 mutants for Qua375. Alternatively, the authors should explain why they changed their control locus.

2. Figure 1f makes it appear that the fraction of eMMEJ deletion events in Qua213 is about one-half

that of the TMEJ events, but Fig 1g shows this not to be the case. Maybe Fig 1f could be modified to show the relative percentages of each type of event, rather than showing 3 or 6 representative events?

Because TMEJ can also create small deletions, perhaps it's more accurate to refer to eMMEJ events as extended deletions (e-dels)? Also, what is the relative e-del frequency for the Qua1277 locus? The prediction is that it should be less than Qua213 because more resection would be needed on both sides.

3. The TMEJ repair events in *helq-1* mutants have longer insertions that appear to be due to multiple switching events during TMEJ. The authors suggest that HELQ is important for extended fill-in synthesis following initial MH annealing by POLQ and in its absence, there is a block to TMEJ that results in more rounds of primer-template switching. If the helicase domain of *C. elegans* POLQ is able to remove proteins (RPA or RAD51) from ssDNA like its mammalian counterpart, then it isn't clear to me why HELQ would be necessary in worms proficient for POLQ helicase activity. Is this phenotype exacerbated (or suppressed?) in worms lacking *polq-1* helicase activity?

4. Figure 4e should include the repair outcomes for the *helq-1 polq-1* double mutant.

5. There are several studies that may be relevant to the authors' models and that they should address in the discussion:

First, two studies in *Drosophila* have investigated roles for the fly HELQ in break repair. One showed that it promotes SSA, but not HR (Johnson-Schlitz et al., 2007; doi.org/10.1371/journal.pgen.0030050), while the other failed to identify a role in either pathway (Wei and Rong, 2007: doi: 10.1534/genetics.107.077693).

Second, while the authors suggest that HELQ might strip proteins from ssDNA, the Warn et al. paper (reference 45) showed that *C. elegans* HELQ can disassemble RAD-51 from dsDNA, but not ssDNA, in vitro. In the current paper, it seemed that

Third, Meier et al., 2021 (<https://doi.org/10.1371/journal.pone.0250291>) propose an alternative model for tandem duplication formation in *helq* mutants, via MM-BIR.

6. The authors often refer to SDSA as 'error-free', but it can be mutagenic (Hicks et al., 2010: doi: 10.1126/science.1191125). Perhaps consider 'high fidelity' or something similar?

7. Small corrections:

- Line 235: should reference Figure 3f?

- Line 301: "...annealing of extensive complementary bases at break ends."

- Line 333: SDSA does require Rad1/10 (XPF/ERCC1) to remove flaps prior to ligation (Ivanov and Haber, 1995; doi: 10.1128/mcb.15.4.2245).

- Line 355: The *Drosophila* genome encodes both HELQ and POLQ and TMEJ events in flies are often complex, with patchwork configurations, in contrast to the plant example.

- Line 374: consider "...HELQ in eMMEJ and POLQ in TMEJ."

We would like to thank all reviewers for their time and support, and for the constructive remarks and suggestions, which helped us to increase the quality and readability of our manuscript.

Reviewer #1 (Remarks to the Author):

Kamp et al argue for a significant role for Helq in every DSB repair pathway except NHEJ. A large number of elegant assays generate some striking data, which taken together mostly support the argument above. It's a mostly beautiful piece of work, and will be of wide interest.

Major concerns that should be addressed:

1) Most importantly, the authors are missing a good argument for gross biological significance. For example, they describe dramatic reductions in aberrations (error prone repair) in double and triple mutants (e.g. helq+polq, helq+polq+lig4) – is there a compensating loss of viability? If not, can they at least show doubles or triples are hypersensitive to damaging agents?

Given the examples mentioned above we have interpreted this comment to pertain to gross biological function for *C. elegans*, to which we comment below. However, if the concern relates to the relevance of HELQ to mammals/human health we would like to refer to the fact that mouse HelQ suppresses tumorigenesis and prevents germ cell attrition (Adelman et al., Nature 2013), and to the best of our knowledge, no homozygous knockout individual has been identified in large scale population studies, arguing for an essential function for human development. Also, HelQ is evolutionarily conserved from archaea to humans, being lost only in some clades (such as in fungi).

With respect to consequences of DNA repair defects for worms, we would like to point out that in the laboratory, these animals, with a genome size 30 times smaller than that of a human or a mouse, are cultured under non-stressed, optimal growth conditions. In contrast, in the natural environment *C. elegans* hibernate as dauer larvae for weeks, move through soil and feed on decaying fruit - conditions (likely) much more toxic to DNA and thus likely providing a selective advantage to animals with a robust DNA damage response.

Our work over the years has indicated that under standard laboratory conditions the number of DNA breaks that require error-prone repair is low. Whole genome sequencing of propagated *C. elegans* strains has revealed a very low level of spontaneous mutagenesis of the type indicative of end joining repair: in wild-type animals we found only a few deletions in over 200 generations of animal growth (Roerink, van Schendel et al. 2014), which is slightly enhanced in Ku80 and Lig4 deficient animals (Kamp, van Schendel et al. 2020), and reduced in PolQ deficient animals (van Schendel, Roerink et al. 2015). The latter may predict loss of germ cells (that would otherwise carry mutations) in PolQ deficient animals, yet only when the level of endogenous substrates was boosted profoundly (>40 fold, e.g. by in addition knocking out TLS polymerase polH and PolK) we were able to detect an obvious effect on animal fitness (Roerink, van Schendel et al. 2014).

Nevertheless, as suggested by this reviewer, animal fitness can also be assayed upon exposure to exogenous stress. In newly performed experiments, we find HelQ deficient animals to be more sensitive to ionizing radiation than wild-type animals (in line with (Muzzini, Plevani et al. 2008)), as is the situation for PolQ deficient animals. We find that animals double deficient for HelQ and PolQ are even more sensitive, in agreement with non-overlapping functions. Lig4 deficiency does not contribute to IR induced sensitivity of germ cells (Kamp, van Schendel et al. 2020).

We have now added these data to the supplementary data.

2) The genetic argument for a role for helq in Polq dependent and independent repair by

TMEJ/MMEJ is good. The authors should perform similar epistasis analysis for SSA (i.e. *helq* with *xpf*) and SDSA (i.e. *helq* with *brc1*).

Concerning epistasis of HELQ-1 with BRC-1 in SDSA:

We have now included SDSA epistasis analysis of *helq-1* with *brd-1*, the interaction partner of *brc-1*; *brd-1* and *brc-1* animals behave identical in all previous experiments (e.g. (Kamp, van Schendel et al. 2020)). We found that the number of cells expressing GFP, as a proxy for HDR, is zero in animals double mutant for *brd-1* and *helq-1*, whereas single mutants still had some residual HDR activity. These data argue for overlapping and non-overlapping roles of BRCA1/BARD1 and HELQ in DNA double-strand break repair via homologous recombination. A similar conclusion can be drawn from the newly included analysis of ionizing radiation sensitivity, in which double mutant animals are more hypersensitive than either of the single mutants, which by themselves are more sensitive to IR-induced damage than wild-type animals.

These data may suggest two modes of HDR that are differently affected by either BRCA1/BARD1 or HELQ deficiency. However, we wish to refrain from overly speculating on potential explanations or implications. For one, similar to the frequently-used DR-GFP assay in mammalian cells, we still have an incomplete understanding of all the ins and outs (e.g. with respect to DSB repair intermediates and outcomes) of the GFP-based assay we use as a proxy for HDR.

But more importantly, to us the jury is still out on the nature of the HR defect in (*C. elegans*) BARD1/BRCA1 mutations: BRCA1/BARD1 is not needed for homologous recombination during meiosis in *C. elegans* germ cells, which has suggested to result from (at least) two modes of repair: i) SDSA leading to non-crossover products, and ii) double Holliday Junction (dHJ)-mediated DSB repair leading, in part, to crossovers. Instead, BARD1/BRCA1 has been suggested to act specifically in an SDSA mechanism that uses the sister chromatid as a repair template. However, to the best of our knowledge, it is currently not known i) whether intersister repair is completely absent in *brc-1/bard-1* animals, ii) whether dHJ-mediated DSB repair is exclusive to germ cells and not also active in *C. elegans* somatic cells, iii) whether the HR defect in *brc-1/brd-1* is a lack of sufficient resection of DSBs, or a lack of invasion of resected ends into the homologous donor (the sister), or a combination thereof, and that some intermediates may funnel into another repair mode in absence of BRCA1/BARD1. All of these possibilities may underlie the residual level of GFP expressing cells in *brc-1/brd-1* animals that apparently rely on HELQ functionality.

We thus feel that we could lead our data in too many different and open directions, yet in our opinion, none of that discussion is central to the HELQ work we here describe.

We have added the new data to the supplementals and remarked on the outcomes minimally in the text of the manuscript .

Concerning epistasis of HELQ-1 with XPF-1 in SSA:

We have performed the suggested experiments and now include those in the supplemental data. The data argue that HELQ-1 and XPF-1 can act non-epistatically in SSA; in fact, *xpf-1* deficiency is not essential for the elevated SSA activity we read out in a *lig-4* background in our reporter system. Also here, in order to provide an explanation that goes beyond mere speculation, we feel that we would need to further research the biology of processing annealed single-strands, and potential redundancies herein, in those (cycling) cells of a developing animal that have highly proficient SSA when NHEJ is impeded.

3) Regarding sites devoid of large microhomologies; the authors initially describe Qua375 as such a site, then in line 155, it is Qua317. Are these different sites (then why use different ones?), or is one a typo?

We wished to thank the reviewer for pointing this out. Over the years we have used multiple Qua loci as controls. To be consistent, we have now performed the experiments depicted in figure 1 for Qua317, and replaced the Qua375 data for the Qua317 data.

4) In Figure 1g, please note the number of deletions scored for each G4 site in the figure (i.e. how many deletions is 100%?; n>360 is too vague).

We have added those numbers to the figure.

5) Line 109, citation 31; please elaborate.

We now elaborate on this issue in the text.

6) Line 183; please quantify what is meant by more frequent “patchwork appearance”, or omit.

We have added the quantification.

7) Line 336; “As previously proposed...” please also cite Jeremy’s work (ref 11) arguing POLQ suppresses a need for larger microhomologies, as well as PMID: 32234782 showing larger microhomologies make POLQ less prone to abortive synthesis/templated insertions. Similarly Line 341; please again cite ref 11 that larger microhomologies eventually make POLQ dispensable for a-EJ in mammals.

We now refer to this work in the discussion section.

8) In the introduction and discussion, please discuss related work in flies (PMC1667076, PMC2013711). The latter is especially relevant, as it argues against a role for HELQ in SSA in flies.

Thanks for pointing this out. Indeed, some seemingly contradictory work exist with regards to a potential role for mus301 (HELQ) in SSA: Earlier work clearly point to a role for mus301 in SSA (Johnson-Schlitz, Flores et al. 2007), but this was unconfirmed in PMC201371. The authors of PMC2013711 (Wei et al.) explain the discrepancy in their discussion: “*JOHNSON-SCHLITZ et al. (2007) tested many of the same mutations that we have tested in this study. The two studies led to consistent results for some genes (i.e., spnA, okr, and lig4), but not others. We did not detect an effect of mei-9, mus101, and mus301 mutations on SSA repair. We suspect that the different results were mainly caused by the difference in the repair assays employed. In our assays, the white duplication is ~3000 bp, which is significantly longer than the 147-bp repeat used by Johnson-Schlitz et al. In addition, the intervening sequence flanked by the repeats is ~1.5 kb in our assay, again significantly longer than the one used in the other study. Either one or both of these features may have rendered our assay insensitive to those mutations tested.*”

The assay used by Johnson-Schlitz et al. is much more similar to our SSA reporter than the assay used in Wei et al., and our outcomes are thus similar.

We now mention and refer to this work in the discussion.

Grammar and style suggestions.

We would like to thank the reviewer for careful reading such that the readability of our work is enhanced

1) Line 55: the manner in which “decent” describes “size” is at best vague, and likely to be confusing to many readers.

We have removed “of decent size”

2) Line 174: “One potential explanation...” has a comma splice that makes the sentence difficult to understand.

We have rephrased this sentence

3) Line 234: “...and in perfect agreement...”, and line 291 “...perfectly fits...”. Suggested that the authors will inspire more agreement amongst readers without offering opinions as to “perfection”.

We have removed perfect and perfectly

4) Line 310 “...besides SDSA...” this clause makes the sentence difficult to parse.

We have removed this clause

5) Line 317 “...in part...” suggest change to “partly”

We have changed this

6) Line 319 “...only to about...” this clause makes the sentence difficult to understand.

We have amended the sentence

Dale Ramsden

Reviewer #2 (Remarks to the Author):

This is a review of NCOMMS-21-13096, which is an elegant study of the influence of the gene HELQ on chromosomal double strand break (DSB) repair and genome stability in *C. elegans* (worms). Importantly, even beyond the mechanistic findings with HELQ per se, the classes of phenotypes observed in HELQ- worms provides significant insight into the classes of backup chromosomal break repair pathways that exist in this organism. It is also a significant study, since the factors that mediate MMEJ/SSA beyond POLQ and RAD52 have remained poorly understood. The study is elegant in that it uses several complementary reporter assays involving distinct types of DSBs: spontaneous events caused by G4 and loss of FANCI, as well as I-SceI and CAS9 DSBs. Several classes of DSB repair events are found to be suppressed/promoted by HELQ that point to a role in events requiring use of longer stretched of homology, which are contrasted from those promoted by DNA Polymerase theta. The manuscript is also written in a compelling manner. For two of the results, however, I would request additional details of the findings, either in the main figure or in supplemental.

1. In Fig 3f, there is a striking increase in templated insertions / insertions in the *lig-4/helq-1* worms. Some examples of these insertions and/or some classification of at least the most common types of insertions will stimulate research in this area. It might also be useful to speculate on the meaning of this result. For examples, this result may point to a role for HELQ in suppressing insertional mutagenesis / suppression of translesion polymerases, and/or it may be that HELQ is important to unwind terminal DSB ends, and in the absence of this activity, polymerases generate ligatable intermediates.

We have adapted the supplemental figure to include all insertions (supplementary figure 2), including their classification, i.e. patchwork insertions, templated insertions and insertions of unknown origin. In the original manuscript, we have speculated on the underlying biology, i.e. that HELQ facilitates efficient TMEJ by removing obstacles for processive action by polymerase theta, thereby suppressing primer-template switching. We now elaborate on the increase of insertions and their nature within the results section, to put more emphasis on this finding. In future work we aim to work towards mechanistic insight into this matter e.g. by developing next-generation sequencing approaches for *C. elegans* to ideally capture thousands of TMEJ junctions, such that we are able to analyse hundreds of these patchwork templated insertions.

2. For the tandem duplications found in Fig 4f, I have a similar comment. Could the authors show some examples of these TDs? How long is the homology involved in the TDs? Finally, some discussion of Ralph Scully's work on how deregulation of end resection can result in long-tract gene conversion may be warranted here, but of course only if the data on the TDs here are informed by that literature. In summary, more description / diagrams of the nature of the TDs will also increase the impact of the study.

We have added all identified tandem duplications in a supplementary excel file. Moreover, we now include a supplementary figure depicting the microhomology and insertions at tandem duplications. As mentioned above at point 1, we need many more data points (and thus novel technology to get to those numbers), to start describing features such length, distance, sequence composition, etc.

While we are following the very elegant work of the Scully lab closely, we did not manage to discuss their work in a way that would not break the flow of our manuscript. Instead, we refer to their work within the discussion section when we discuss TD aetiology.

Reviewer #3 (Remarks to the Author):

In this paper, Kamp et al. investigate the source of residual microhomology-mediated end joining that occurs in *C. elegans* polq-1 mutants. They identify the DNA helicase HELQ as being essential for MMEJ repair of G4-related DSBs with flanking microhomologies of greater than 20 nts (which they call eMMEJ). In addition, they show that HELQ facilitates TMEJ and in its absence, repair junctions have larger and more complex insertions. Using several reporter assays, they establish that HELQ-1 is needed for SSA in a lig4 background and for efficient SDSA. In its absence, tandem duplications accumulate in the genome during mutation accumulation experiments. Overall, their results suggest an important role for HELQ in double-strand break repair that involves annealing of complementary ssDNA molecules greater than 20 nt in length.

The novelty of this study lies in its identification of a protein important for the small fraction of end-joining repair events that occur in the absence of pol theta. These have previously been observed in worms, flies, and mouse cells, but their etiology has remained unknown. The involvement of HELQ in SSA and SDSA has been suggested in other systems, but this work brings these observations together under the umbrella of a postulated role of HELQ-1 in removing inhibitory proteins from ssDNA prior to completion of break repair. The data are clear and the figures nicely convey the main points. While I have some questions about the interpretation of some of the data, I think that the study as a whole provides significant insight into DSB repair and pathway choice in *C. elegans* (but perhaps not in other metazoans, see below).

Points to consider:

1. It isn't clear why different control sequences are used for Figure 1 (Qua375) and Figure 2 (Qua317). It would be useful to include sequencing data for the Qua317 locus in the different mutant backgrounds, or alternatively show the sequencing data in helq-1 and dog-1 polq-1 helq-1 mutants for Qua375. Alternatively, the authors should explain why they changed their control locus.

We wished to thank the reviewer for pointing this out. Over the years we have used multiple Qua loci as controls. To be consistent, we have now performed the experiments depicted in figure 1 for Qua317, and replaced the Qua375 data for the Qua317 data.

2. Figure 1f makes it appear that the fraction of eMMEJ deletion events in Qua213 is about one-half that of the TMEJ events, but Fig 1g shows this not to be the case. Maybe Fig 1f could be modified to show the relative percentages of each type of event, rather than showing 3 or 6 representative events?

We agree with this point. We amended the figure, which primary purpose is to highlight the preferred usage of the microhomologous sequences for repair in the respective genetic backgrounds, to be more in line with the frequencies of deletion formation: i) for Qua917 we included the percentage of deletions that had eMH usage, ii) we changed the number of deletions per genotype to have it more aligned with the detected frequency, iii) we included the actual numbers in Figure 1G.

Because TMEJ can also create small deletions, perhaps it's more accurate to refer to eMMEJ events as extended deletions (e-dels)? Also, what is the relative e-del frequency for the Qua1277 locus? The prediction is that it should be less than Qua213 because more resection would be needed on both sides.

We thank the reviewer for this suggestion but we chose not to change the nomenclature. One reason is that the outcomes of TMEJ and eMMEJ in genetically unperturbed conditions (can) overlap, which will run into difficulty as name giving will then be subjected to interpretation. Another reason concerns terminology and definitions:

In general, we give word choice a great deal of attention in trying to best describe our data. With respect to pathway definition in DSB repair we feel issues inevitably arise and have also previously arisen: defining pathways by their outcomes can be problematic because of the overlap - NHEJ, TMEJ and eMMEJ can all produce the same outcome on DSBs of a specific sequence context. Alternatively, defining a pathway by its genetic requirements can also be problematic as some proteins can have roles in different pathways (*i.e.* HelQ activity in SSA, SDSA, HDR and TMEJ). In cases where a specific factor is quintessential to biology that can be well described, and can be separated on the basis of those genetic requirements from other pathways that produce similar outcomes, we feel that it is more clear to use the name instigated by the protein(s). Hence, we previously coined the term TMEJ for all cases where we have shown Pol theta necessity, instead of using more umbrella terms like altEJ or MMEJ. We felt that the earlier name synthesis-dependent MMEJ (SD-MMEJ), coined by the McVey lab, which highlights mechanistic features is/was also well chosen, however, also here one could argue that it may be an umbrella term as one can envisage Pol theta-independent SD-MMEJ, but more importantly: 20-50% of TMEJ outcomes in *C. elegans* are not defined by MH at the junction; it is in those cases not obvious that the products are microhomology-mediated (MM), while these were demonstrated to be Pol theta-mediated (TM).

Given our wish to be precise yet also reserved with respect to introducing new terminology we reluctantly have termed the EJ activity that is independent on Lig4 and PolQ, but clearly categorized by extensive MH at the junction, as eMMEJ. Yet, we do not advocate the use of this term more frequently in the future: as written in the discussion section, we envisage that TMEJ, eMMEJ and SSA are not necessarily different "pathways" but instead all describe context-dependent action of a sequence-based way of joining the ends of a DSB (in contrast to cNHEJ which is grosso-modo not sequence- but protein-based).

We calculated the deletion frequency of Qua1277 and indeed found that to be lower than the deletion frequency at Qua213. We now include the frequencies in Figure 1g.

While we have now added frequencies to figure 1 we do not wish to overstate potential conclusions based on these frequencies, as the assay is based on nested PCRs and possibly influenced by

sequence context and G4 configuration.

3. The TMEJ repair events in *helq-1* mutants have longer insertions that appear to be due to multiple switching events during TMEJ. The authors suggest that HELQ is important for extended fill-in synthesis following initial MH annealing by POLQ and in its absence, there is a block to TMEJ that results in more rounds of primer-template switching. If the helicase domain of *C. elegans* POLQ is able to remove proteins (RPA or RAD51) from ssDNA like its mammalian counterpart, then it isn't clear to me why HELQ would be necessary in worms proficient for POLQ helicase activity. Is this phenotype exacerbated (or suppressed?) in worms lacking *polq-1* helicase activity?

We agree: at present it is also unclear to us in what way TMEJ is facilitated by HELQ action given that a similar activity lies within the POLQ protein.

We have now added to the discussion section the hypothesis that eMMEJ may on occasion act downstream of aborted TMEJ. In cases where TMEJ has already started DNA synthesis but cannot be completed (for yet unknown reasons) one could envision an intermediate that can also be resolved by eMMEJ, because the newly synthesized DNA stretch is complementary to DNA at the other break end. If HELQ-driven eMMEJ acts to repair this situation, a "simple" junction would result (the DNA tract synthesized in the TMEJ reaction will not show up in the product as it will be used to anneal both ends). However, in the absence of HELQ, cells will rely on TMEJ to repair this intermediate, which may preferentially use the outer ends of the 3' overhangs. As a consequence templated insertions arise.

With respect to the experimental suggestion: we have previously generated worms with a mutated allele of *polq-1* which should produce a protein that in other systems have shown to affect helicase activity. These worms behaved similar to *polq-1* null mutant animals in our assays, being completely deficient in TMEJ, which may support a conclusion that the helicase of *C. elegans* POLQ is vital in TMEJ. However, because we have no information concerning functionality of the mutated protein *in vivo* (folding, recruitment, polymerase action) we would like to refrain from making too bold statements.

4. Figure 4e should include the repair outcomes for the *helq-1 polq-1* double mutant.

We have performed the suggested experiments and have included the data in the new version of the manuscript: In line with the hypothesis that TMEJ is responsible for the vast majority of mutagenic end-joining in the *C. elegans* germline (van Schendel, Roerink et al. 2015) we exclusively found HDR events in *polq-1* mutant animals, and in agreement with HELQ being essential for HDR in *C. elegans* we did not retrieve any converted alleles in *helq-1 polq-1* double mutant animals.

5. There are several studies that may be relevant to the authors' models and that they should address in the discussion:

First, two studies in *Drosophila* have investigated roles for the fly HELQ in break repair. One showed that it promotes SSA, but not HR (Johnson-Schlitz et al., 2007; doi.org/10.1371/journal.pgen.0030050), while the other failed to identify a role in either pathway (Wei and Rong, 2007: doi: 10.1534/genetics.107.077693).

We thank the reviewer for pointing those out and we have now incorporated this work in the manuscript.

Second, while the authors suggest that HELQ might strip proteins from ssDNA, the Warn et al. paper

(reference 45) showed that *C. elegans* HELQ can disassemble RAD-51 from dsDNA, but not ssDNA, in vitro.

We agree. In favour of our suggested explanation, additional (also recent) *in vitro* work exist which shows that archaeal and human HelQ can interact with RPA and displace it from ssDNA. That work is now discussed.

Concerning the notion that purified *C. elegans* HELQ-1 has been shown to remove RAD51 from dsDNA, we wished to note that the same lab demonstrates that “HELQ possesses an intrinsic ability to capture RPA-bound DNA strands and then displace RPA to facilitate annealing of complementary strands” (<https://www.researchsquare.com/article/rs-583248/v1>). Because this work is not yet published we felt uncomfortable referring to this work in our manuscript.

Third, Meier et al., 2021 (<https://doi.org/10.1371/journal.pone.0250291>) propose an alternative model for tandem duplication formation in *helq* mutants, via MM-BIR.

We have now included a reference to this manuscript, but chose to not discuss their model as it is speculative and not supported by experimental evidence. To elaborate on this issue, the presented model, in particular invoking perturbed replication as a source for tandem duplication, is based on the notion that tandem duplications are found nearby sequence repeats. That manuscript, however, does not address whether TDs are preferentially found in the vicinity of repeats. We now have addressed this issue for their data set as well as ours and while we have confirmed their numbers in analysing their data, and also in our data found that ~ 50% of TDs are located nearby sequence repeats, we find the same 50% if we analyse sets of random, computationally-generated tandem duplication. This argues that there is no support for a causal involvement of repeated sequence, hence also no circumstantial evidence for the model.

6. The authors often free to SDSA as ‘error-free’, but it can be mutagenic (Hicks et al., 2010: doi: 10.1126/science.1191125). Perhaps consider ‘high fidelity’ or something similar?

We have removed ‘error-free’ or replaced it with ‘high fidelity’ at relevant positions in the text.

7. Small corrections:

- Line 235: should reference Figure 3f?

We have now corrected this

- Line 301: “...annealing of extensive complementary bases at break ends.”

We have now corrected this

- Line 333: SDSA does require Rad1/10 (XPF/ERCC1) to remove flaps prior to ligation (Ivanov and Haber, 1995; doi: 10.1128/mcb.15.4.2245).

We have now removed this clause

- Line 355: The *Drosophila* genome encodes both HELQ and POLQ and TMEJ events in flies are often complex, with patchwork configurations, in contrast to the plant example.

We follow the work, mostly of the McVey lab, on this issue closely and recognise that also in *Drosophila* templated insertions have been found that have a patchwork configuration. Notably, also in *C. elegans* we previously found that a percentage of footprints upon repair of transposon-induced

DSB (roughly 1 in 5 templated insertions) are of a patchwork configuration (van Schendel et al., Nat. Commun. 2015). However, in both worms and flies the majority of repair products are simple deletions (e.g. in flies: ~ 75% in lig4 deficient conditions (khodaverdian et al., NAR 2017). In contrast, in the plant *Arabidopsis thaliana*, the majority are non-simple, and most cases are of a complex configuration. We thus feel that the fly and worm data are more alike, but because we realise it is difficult to compare (still limited) data between species, in which DSB-context may also be of importance, we chose to limit ourselves by only pointing to the plant data.

- Line 374: consider "...HELQ in eMMEJ and POLQ in TMEJ."

We have implemented this correction

References

- Johnson-Schlitz, D. M., C. Flores and W. R. Engels (2007). "Multiple-pathway analysis of double-strand break repair mutations in *Drosophila*." PLoS Genet **3**(4): e50.
- Kamp, J. A., R. van Schendel, I. W. Dilweg and M. Tijsterman (2020). "BRCA1-associated structural variations are a consequence of polymerase theta-mediated end-joining." Nat Commun **11**(1): 3615.
- Muzzini, D. M., P. Plevani, S. J. Boulton, G. Cassata and F. Marini (2008). "Caenorhabditis elegans POLQ-1 and HEL-308 function in two distinct DNA interstrand cross-link repair pathways." DNA Repair (Amst) **7**(6): 941-950.
- Roerink, S. F., R. van Schendel and M. Tijsterman (2014). "Polymerase theta-mediated end joining of replication-associated DNA breaks in *C. elegans*." Genome Res **24**(6): 954-962.
- van Schendel, R., S. F. Roerink, V. Portegijs, S. van den Heuvel and M. Tijsterman (2015). "Polymerase Theta is a key driver of genome evolution and of CRISPR/Cas9-mediated mutagenesis." Nat Commun **6**: 7394.

Reviewers' Comments:

Reviewer #1:

Remarks to the Author:

The authors make a strong case against excessive speculation on the results of *brd1* and *helq* epistasis. However, when referring to the XPF independency of their SSA assay (line 210), it would probably be helpful to readers not familiar with past work to comment here that this is at least a little surprising (in that it is inconsistent with results in e.g. budding yeast; the latter point is only mentioned in the intro).

Reviewer #2:

Remarks to the Author:

I am pleased to report that the authors have added additional information in supplemental files and edits to the text to clarify the results in this impactful study. Thus, my concerns have been adequately addressed.

Reviewer #3:

Remarks to the Author:

The authors have sufficiently responded to all my concerns with new experimental data and a more nuanced discussion of eMMEJ and its relationship to TMEJ. I appreciate their attempt to contextualize their findings in other organisms that do and do not have HelQ. I think that this work represents an important advance in the field and will be well-received.

REVIEWERS' COMMENTS

Reviewer #1 (Remarks to the Author):

The authors make a strong case against excessive speculation on the results of brd1 and helq epistasis. However, when referring to the XPF independency of their SSA assay (line 210), it would probably be helpful to readers not familiar with past work to comment here that this is at least a little surprising (in that it is inconsistent with results in e.g. budding yeast; the latter point is only mentioned in the intro).

>> We amended the text accordingly

Reviewer #2 (Remarks to the Author):

I am pleased to report that the authors have added additional information in supplemental files and edits to the text to clarify the results in this impactful study. Thus, my concerns have been adequately addressed.

>> No further action is needed

Reviewer #3 (Remarks to the Author):

The authors have sufficiently responded to all my concerns with new experimental data and a more nuanced discussion of eMMEJ and its relationship to TMEJ. I appreciate their attempt to contextualize their findings in other organisms that do and do not have HelQ. I think that this work represents an important advance in the field and will be well-received.

>> No further action is needed